# Providing a Visual Understanding of Holography Through Phase Space Representations

**Tobias Birnbaum** [1,2,*] **, Tomasz Kozacki** [3] **and Peter Schelkens** [1,2]

1    Department of Electronics and Informatics (ETRO), Vrije Universiteit Brussel (VUB), Brussels B-1050, Belgium; pschelke@etrovub.be

2    imec, Department of Digital & User Centric Solutions (DUCS), B-3001 Leuven, Belgium

3    Department of Mechatronics, Warsaw University of Technology, 02-525 Warsaw, Poland; t.kozacki@mchtr.pw.edu.pl

*    Correspondence: tbirnbau@etrovub.be



**Featured Application: Visually interpretable phase space representations can aid the development of new methodologies in digital holography in general and can be evaluated for any digital hologram. Concrete applications presented in the paper are: coarse scene depth estimation, scene analysis, and digital hologram (processing) artifact analysis.**

**Abstract:**   Digital holograms are a prime example for signals, which are best understood in phase space—the joint space of spatial coordinates and spatial frequencies. Many characteristics, as well as optical operations can be visualized therein with so called phase space representations (PSRs). However, literature relies often only on symbolic PSRs or on, in practice, visually insufficient PSRs like the Wigner–Ville representation. In this tutorial-style paper, we will showcase the S-method, which is both a PSR that can be calculated directly from any given signal, and that allows for a clear visual interpretation. We will highlight the power of space-frequency analysis in digital holography, explain why this specific PSR is recommended, discuss a broad range of basic operations, and briefly overview several interesting practical questions in digital holography.

**Dataset:** www.erc-interfere.eu

**Dataset License:** CC-BY-NC-SA

**Keywords:**   digital holography; phase space representation; time-frequency representation; Wigner–Ville; spectrogram; S-method

## 1. Introduction

The phase space of a signal is the joint space of the signals native domain and its Fourier domain. It can consist of time and frequency for temporal signals or spatial positions and spatial frequencies for spatial signals. Phase space representations (PSRs; in signal processing literature the term time-frequency representations is more common) are an essential tool initially proposed to aid phase space analysis, which tries to understand systems whose frequency spectrum changes over the course of the signal by visualizing quasi-instantaneous frequencies as they change. They quickly proved themselves useful in many applications of science and engineering. Examples are: incoherent optics [1], biomedical engineering [2,3], radar [4,5]/sonar/seismic wave-processing [6], telecommunications [7], power generation [7], combustion analysis [7], machine-condition monitoring [7], geophysics [7], or quantum mechanics [7]. When PSRs are applied to temporal signals and are equipped with a logarithmic decrease in frequency resolution,

favoring frequency over time resolution at low-frequencies, they are used in auditory applications such as instrument or musical score identification [8], or even reconstruction of long lost antiquities as for example Brahms Hungarian Rhapsody [9]. Even in the recent past, PSRs continue to provide vital, new insights in research, such as the S-method applied to gravitational waves [10].

In digital holography (DH) the Fourier conjugated variables are usually spatial position $\xi, \eta$ and spatial frequency $f_\xi, f_\eta$, loosely speaking the number of fringe oscillations per unit of space. To avoid confusion, we shall continue to refer to the joint space-frequency domain as phase space, and representations thereof as PSRs. Several authors have recognized the potential of phase space discussions in DH, since point-spread functions $P$ (PSFs) are a prime example of signals whose frequency distribution varies with space. Figure 1a shows the real parts of two exemplary PSFs, recorded at some distance $z$ along the optical axis in an off-axis position denoted by $\Delta x$. Note, the oscillations in the spatial domain are more rapid the further off-axis the recording is considered. We will elaborate in more depth on this later on.

In Figure 1b we provide an overview of how to use PSRs to perform phase space analysis on DH and over the involved dimensions. Although 2D holograms span a 4D phase space, it will be often sufficient for analysis to consider only 2D cross-sections of phase space obtained from 1D cross-sections of the hologram. Most of this paper will use horizontal or vertical cross-sections centered on the hologram.

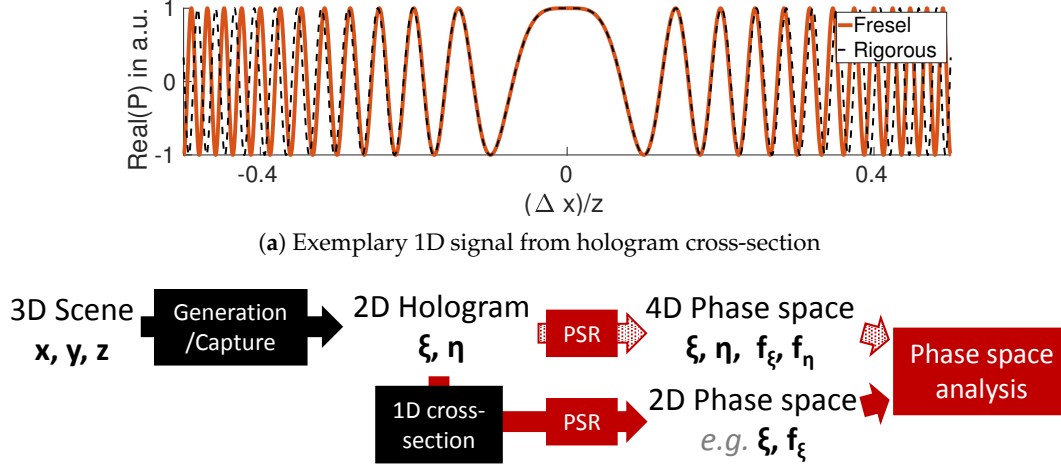

(**a**) Exemplary 1D signal from hologram cross-section

(**b**) Dimensionality of phase space analysis in digital holography

**Figure 1.** (**a**) Real part of a 1D point-spread function of a spherical (dashed, black; Rigorous form) and a quadratic wavefront (continuous, red; Fresnel approx.). (**b**) Application of phase space representations (PSR) to digital holograms.

The majority of previously published work about the application of PSRs to DH is based on the Wigner–Ville (WV) distribution (Section 9.6 in [11]) and the short-term Fourier transform (STFT) (Section 9.2.6 in [11]). These are vital for performing calculations, as they are invertible; however, upon visualizing phase space they suffer from substantial artifacts as we shall see later on. Henceforth, most of the time only qualitative WV charts are used for visualization, which are not as useful to practitioners confronted with imperfect, partially unknown, or processed data.

In science, the intuition for finding a solution to a problem is often guided by appropriate visualizations, therefore we propose the use of the S-method [12] for visual exploration of phase space in DH and performing a diagnostic of processing artifacts that are not easily visible otherwise. For this, it is sufficient to evaluate it on individual 1D slices of 2D holograms even for deep 3D scenes (see Figure 1b). We show that this representation stands out from many other PSRs by being fast to calculate, generic, and allowing for a good visual interpretation.

Aided by this PSR, we discuss a variety of different applications in DH—some of which may only be most easily addressed with visual interpretations. By doing so, we hope to raise awareness of the reader to this valuable tool in the context of DH. Furthermore, we aim to provide a concise collection of all relevant details required to perform this analysis.

After a short literature review, we will motivate the use of phase space analysis for DH in general in Section 2. Section 3 will then introduce visually interpretable PSRs and outline the S-method. In Section 4, we utilize PSRs to discuss the interference of object and reference wavefields. The main part of our paper will discuss different applications of PSRs to DH in Section 5. We will briefly explain in Section 5.1 what one can learn through phase space analysis about a given macroscopic DH, for which no knowledge on its scene composition is available. Thereafter, we elaborate more on details concerning the space-bandwidth product of DHs in Section 5.2 and on scene depth estimation in Section 5.3. In Section 5.4 the S-method is demonstrated in the analysis of DH processing artifacts. The effects of common operations applied to DHs are provided in Section 5.5. In Section 5.6, we show how the degree of surface roughness influences the PSR footprint of a DH. We close with an outlook to three advanced applications where visually interpretable PSRs aid conceptual understanding. Namely, we touch on multi-object holographic scene segmentation (Section 5.7), perceptual quality assessment of DH (Section 5.8), and provide some thoughts on design and understanding limitations of DH setups (Section 5.9). We conclude our work in Section 6.

This paper is predominantly written as a tutorial for researchers that are unfamiliar with thinking about DH in terms of phase space analysis. Any expert reader, interested only in the novelties, might skip ahead and read about the S-method, newly proposed for DH analysis, in Sections 3.3–3.5, as well as about advantageous applications thereof in Sections 5.3–5.9 .

### 1.1. Related Work

The most renown work on phase space analysis in DH is certainly the book by Goodman on Fourier optics [13] discussing phase space implicitly, the diffraction equation, the space-bandwidth product, the form and shape of exemplary diffraction patterns stemming from various apertures and gratings, as well as a definition of instantaneous frequency based on the Wigner–Ville (WV) distribution. An even more comprehensive book on phase space optics was published by Testorf et al. [14], emphasizing the importance of phase space methods in the entire optics domain, and proposing these methods as unifying modality to capture light propagation within any of the common light models may it be rays, scalar waves, or Gaussian beams. A wide range of applications are discussed in detail to that end. [15] issued another investigation into phase space methods for DH as well as light field and integral imaging. Other authors make use of the phase space methods in order to: perform 3D triangulation of sparsely scattering scenes from 1D DHs [15,16]; devise view-dependent [17,18] or general compression schemes [19] for DH; speed up the generation of holograms [20]; derive generalized sampling theorems [21]; better understand optical operations in DH [22,23]; understand the optical bandwidth of a system [24]. In [25,26], different PSRs were studied for DH microscopy of extremely sparse scenes. These works might be thought of as a predecessor to the current work, where we present a general discussion – applicable also for very dense and macroscopic scenes. We omit any works on temporally resolved holography as therein temporal dynamics are analyzed, whilst we focus on the spatial dimensions and frequencies of static holograms.

Several contributions by the authors have made use of phase space methods, too, e.g., for compression [27–29]; segmentation of holograms with multiple occluding objects [30]; faster computer generation [31]; development of more efficient propagation schemes [32]; comparison of angular and spatial multiplexed spatial-light modulator setups [33] and the study of their perceptual quality [34]; or to design orthoscopic display setups [35]. Despite this certainly non-exhaustive list of references it is evident how important phase space analysis and PSRs are in optics in general and in DH especially.

## 2. Point-Spread Functions in Phase Space

Now, we introduce the fundamental concepts of DH required for the subsequent phase space analysis. The diffraction pattern, recorded on a given manifold, for example a (detector-)plane, generated by a single point-source is called point-spread function (PSF) $P$. A vast amount ($> 10^5$) of super-positioned PSFs, each from a different illuminated point-source in the scene, is what we refer to as a hologram $H(\xi, \eta, 0)$. Within scalar diffraction theory PSFs are complex-valued and described in every point of the recording manifold by one amplitude and one phase. Given an amplitude-phase distribution $h(x, y, z) \in \mathbb{C}$ of point-sources located between the $(x, y, z = const)$ planes with $z \in [z_1, z_2]$, $H$ can be calculated via the first Rayleigh–Sommerfeld integral solution as

$$H(\xi, \eta, 0) = \overline{E_R}(\xi, \eta, 0) \underbrace{\int_{z_1}^{z_2} \int_{-\infty}^{\infty} \int_{-\infty}^{\infty} h(x, y, z) P(\xi, \eta, 0; x, y, z) dx dy dz}_{O(\xi, \eta, 0)}, \tag{1}$$

whereas $\overline{E_R}$ is a conjugated reference wave used for recording. $O$ is called object wave and this point-wise multiplication of two complex-valued wavefields is called interference. Interference is vital in practical setups. For the sake of clarity, we will split the discussion of the phase space of the object and reference waves. The main part of this paper will be concerned only with the object wavefield $O$. Reference waves and the interplay of both waves will be covered in Section 4. For now, we shall assume a plane wave with phase 0, i.e., $E_R = 1$. Given the complex-valued, continuous object wavefield on a surface, a digital hologram can be obtained through pixelation [36]—a sampling with a given pixel pitch $p$ and discretization by some bit-depth. A complex-valued matrix of size $N_\eta \times N_\xi$ is obtained where $\xi$ enumerates the columns.

For a PSF, the spatial frequency distribution, i.e., the way spatial frequencies vary with space, encodes the position of the corresponding point-source in scene space and its relative phase offset. We visualized the dependence on lateral and longitudinal positioning in Figure 2 for two different expressions of the PSFs.

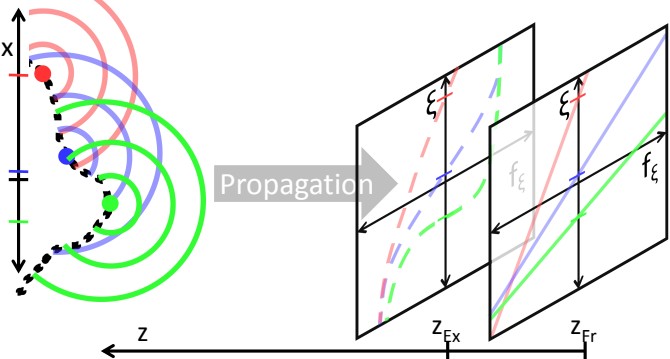

**Figure 2.** 1D hologram genesis starting from 3 specific points, and phase space footprint in the rigorous form at $z_{Ex}$ and within the Fresnel approximation at $z_{Fr}$.

The two most common PSF expressions are the rigorous form, which corresponds to working with spherical wavefronts, and the Fresnel approximation, corresponding to quadratic wavefronts. Rarely, a more crude PSF approximation finds use as well which is called the Fraunhofer approximation. These approximations are used to simplify analysis and substantially speedup calculations as through Equation (1) a different diffraction regime, i.e., a way of (approximate) light propagation, is associated to each of the specific PSF shapes.

One way to classify diffraction regimes is by the Fresnel number $N_{Fr}$, see [13], defined as

$$N_{Fr}(r) := \frac{(0.5 \; \varnothing(\text{aperture}))^2}{\lambda \, |z|} = \frac{0.5 \max(|\xi - x|) \cdot 0.5 \max(|\eta - y|)}{\lambda \, |z|} . \tag{2}$$

If $N_{Fr} > 1$ the PSF takes the rigorous form and its Fourier transform, the phase accommodation kernel of the angular spectrum method, is typically used for propagation of holograms in this regime. If $N_{Fr} \lesssim 1$ the Fresnel approximation is sufficiently accurate for propagation and if $N_{Fr} \ll 1$ propagation can be approximated even further by the Fraunhofer approximation. The reminder of this section will be devoted to analyzing the phase space behavior of these PSF forms.

*2.1. Rigorous Form*

Let $x, y, z$ be a right-handed Cartesian coordinate system in scene space and let $\xi, \eta$ be spatial coordinates of the hologram plane located in $z = 0$, such that $\xi, \eta$ are parallel to $x, y$, respectively and the $z$ axis is aligned with the optical axis of our system. Without loss of generality let the scene be located at $z > \lambda \gg 0$ where $\lambda$ is the wavelength of the illuminating light. Then the PSF $P(r) = P(\xi, \eta, z; x, y, 0)$ of a spherical wave diverging from $(x, y, z)$ towards $z = 0$ is given as

$$P(r) := \frac{z e^{i\varphi(r)}}{i\lambda r^2} \quad \text{with} \quad \varphi(r) := -\frac{2\pi}{\lambda} r \quad \text{and} \quad r := \sqrt{(\xi - x)^2 + (\eta - y)^2 + (0 - z)^2} . \tag{3}$$

$\varphi$ is its (instantaneous) phase and can be used for the definition of its (instantaneous) frequencies $f_\xi$ and $f_\eta$.

$$f_\xi(r) := \frac{1}{2\pi} \frac{\partial \varphi}{\partial \xi} = -\frac{(\xi - x)}{\lambda r} \quad \text{and} \quad f_\eta(r) := \frac{1}{2\pi} \frac{\partial \varphi}{\partial \eta} = -\frac{(\eta - y)}{\lambda r} , \tag{4}$$

with $r$ given by Equation (3). This definition is based on the principle of stationary phase, which states that the (instantaneous) phase $\varphi$, Equation (3) is approximately sinusoidal while varying $\xi, \eta$ over several $\lambda$. It is worth mentioning, that other definitions for instantaneous frequency, which are not based on the principle of stationary phase, do exist. One example is the definition based on marginals evaluated for the WV PSR [14] or higher order derivatives of the phase functions. A detailed list is provided in [37].

Irrespective of the chosen definition, the spatial frequencies of a monochromatic wave may not exceed $\pm 1/\lambda$. For a DH, it furthermore may not exceed $(2p)^{-1}$. Equation (4) describes the rigorous mapping from space $\xi, \eta$ to frequency $f_\xi, f_\eta$ for an ideal PSF without any aberrations. Its phase space is shown in Figure 3c for a 1D PSF, i.e. with $\Delta x := (\xi - x)$ and $r := \sqrt{(\Delta x)^2 + (0 - z)^2}$. The real part of the same PSF, which is shown in spatial domain in Figure 1a, is presented in Fourier space in Figure 3a. Its instantaneous phase function in spatial domain is depicted in Figure 3b.

Spatial frequencies are linked to diffraction angles by the grating equation. For example a discretized $f_\xi$ determines the angle $\theta$ in the $\xi - z$ plane

$$\sin(\theta) = \lambda f_n \quad \text{with} \quad f_n = \frac{n}{Np} \quad \text{and} \quad n \in [-N/2 + 1, N/2] \tag{5}$$

where $N$ is the number of pixels of the hologram along $\xi$ and $p$ is its pixel pitch.

Let us now study the effects of translations along $x, z$ on the instantaneous frequency $f_\xi$. Translations along $x$ correspond directly to translations of $f_\xi$ along $\xi$ as $f_\xi = 0 \Leftrightarrow \xi = x$. Translations along the optical axis $z$ change the steepness of the frequency transition. As $z$ appears only in the denominator for $z \to \infty$ the change of frequencies will happen imperceptibly slow, i.e., $\frac{\partial f_\xi}{\partial \xi}\big|_{\xi = x} \approx 0$. For $z \approx 0$ the step function is obtained. If $z < 0$ then $\xi$ and $x$ trade places and thereby induce a sign change of $f_\xi$. If as shown, a down-chirp (decreasing frequency) is present, the recorded wave is diverging from a point-source located at $z > 0$ towards $z = 0$. A spherical wave diverging

from $z < 0$ towards $z = 0$ is approximated by up-chirps. In Figure 2 we color-coded the footprints of the PSFs belonging to 3 different point-sources.

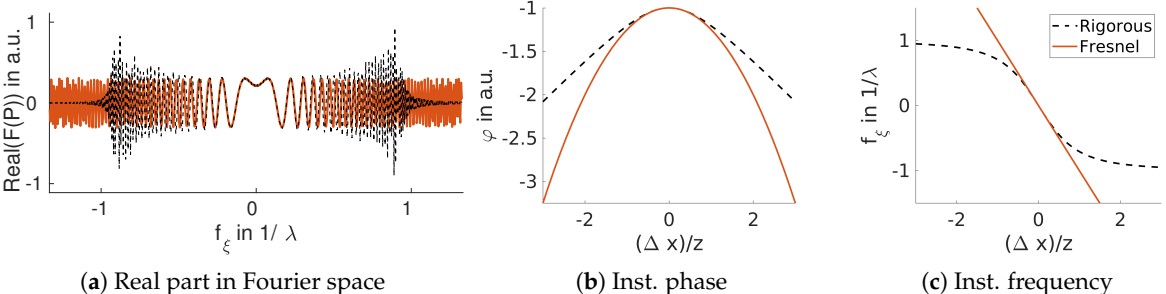

(**a**) Real part in Fourier space　　　　(**b**) Inst. phase　　　　(**c**) Inst. frequency

**Figure 3.** A comparison of the Fresnel approximation with the rigorous form of a single 1D point-spread function in terms of: (**a**) real part in Fourier space; (**b**) instantaneous phase in space; (**c**) instantaneous frequency in space-frequency domain.

### 2.2. Fresnel Approximation

Within the Fresnel approximation each spherical wavefront is approximated as a parabola

$$P_{\text{Fr}}(r) := \frac{1}{i\lambda z} e^{i\varphi_{\text{Fr}}(r)} \quad \text{with} \quad \varphi_{\text{Fr}}(r) := -\frac{2\pi}{\lambda}\left(z + \frac{r^2 - z^2}{2z}\right). \tag{6}$$

Instantaneous phase and instantaneous frequencies are given by

$$f_\xi(r) := \frac{1}{2\pi}\frac{\partial \varphi_{\text{Fr}}(r)}{\partial \xi} = -\frac{(\xi - x)}{\lambda z} \quad \text{and} \quad f_\eta(r) := \frac{1}{2\pi}\frac{\partial \varphi_{\text{Fr}}(r)}{\partial \eta} = -\frac{(\eta - y)}{\lambda z}. \tag{7}$$

As a result that a quadratic instantaneous phase yields a linear instantaneous frequency, the latter can grow unbound in the Fresnel approximation whenever $|\xi - x| \gtrsim z$. In Figure 3a we can observe nicely how the spectrum of a parabolic wavefront extends across the entire Fourier domain for any $|z| < \infty$.

Note, how the Fresnel approximation is hard to grasp for PSFs from either their spatial (Figure 1a) or frequency profiles (Figure 3a) alone. When both domains are jointly regarded in Figure 3c, the nature of the approximation is immediately revealed. Its instantaneous frequency is a 1st order (linear) Taylor expansion to the rigorous PSF around the point $f_\xi = 0$ and gives rise to quadratic wavefronts. It also becomes immediately evident under which conditions the Fresnel approximation holds. Loosely speaking, it holds when the maximal frequency in the hologram is smaller than $1/\lambda$ and that $\max(\xi - x)/z$ and $\max(\eta - y)/z$ are small. This is at the heart of the requirement $N_F \leq 1$.

### 2.3. Fraunhofer Approximation

The Fraunhofer approximation is more crude than the Fresnel approximation, as it approximates the collection of all spherical wavefronts in a scene by a single parabolic wavefront. With a far-field $z \gg 2\pi/(2\lambda)\max(|\xi^2 + \eta^2|)$ we find from Equation (6) for the instantaneous phase

$$P_{\text{Fh}}(r) := \frac{1}{i\lambda z} e^{i\varphi_{\text{Fh}}(r)} \quad \text{with} \quad \varphi_{\text{Fh}}(r) := -\frac{2\pi}{\lambda}\left(z + \frac{x^2 + y^2 - 2x\xi - 2y\eta}{2z}\right) \tag{8}$$

and for the instantaneous frequency therefore

$$f_\xi(r) := \frac{1}{2\pi}\frac{\partial \varphi_{\text{Fh}}(r)}{\partial \xi} = \frac{x}{\lambda z} \quad \text{and} \quad f_\eta(r) := \frac{1}{2\pi}\frac{\partial \varphi_{\text{Fh}}(r)}{\partial \eta} = \frac{y}{\lambda z}. \tag{9}$$

The phase space footprint will be in shape identical to that of the Fresnel approximation in Figure 3 for $\xi = 0$ fixed. Note, the Fraunhofer approximation may also be obtained in the focal plane of a lens [38].

## 3. Explicit Calculation of Phase Space Representations of Arbitrary Digital Holograms

Thus far we have regarded individual point-sources and used precise instantaneous frequency laws to describe their phase space behavior. However, in physical recordings, where the exact positions of point-sources are unknown and aberrations can occur, for DH after a long processing pipeline which can introduce artifacts, or for holograms of large and dense objects, which consists of $\geq 10^5$ PSFs, this approach is no longer feasible. Instead of mapping out the occupied phase space volume PSF by PSF for each point-source, we will analyze the phase space of the fully assembled hologram. Although, phase space analysis can be used for different purposes as well, we will regard only visually interpretable PSRs in this paper, as they can provide great intuition to problem solving.

### 3.1. Selecting the Right Phase Space Representation

There exist many PSRs, which is explained by the fact that it is impossible to craft one representation that can precisely analyze instantaneous frequencies at every point in space due to the Heisenberg–Pauli–Weyl uncertainty principle [39]. Henceforth, there are various trade-offs to be made with respect to properties that shall be retained, such as: visual interpretability, positivity, energy conservation, covariance with space and frequency, invertibility, degree of adaptivity, and many more. We will restrict ourselves to Cohen's class [40], which is the general set of bilinear PSRs that are invariant under space-frequency translations and encloses all off the most frequently used PSRs.

Most of the references cited earlier on, chose the Wigner–Ville (WV) PSR, because it offers a variety of nice mathematical properties such as invertibility, compatibility with linear filtering and modulations, conservation of scalar product and instantaneous frequency (in terms of marginals), and invertibility. Put differently, it lends itself naturally to many mathematical derivations and offers perfect localization ("best resolution") for any signal described by a single, straight line in phase space; such as: pure sinusoids, impulses, or linear frequency chirps. However, visual interpretation becomes unfeasible for signals described in phase space by either multiple lines and/or arbitrary curves because of the introduction of interference terms and their specific placement.

This is illustrated by two examples shown in Figure 4 where several PSRs are evaluated on the central horizontal cross-section of 2D holograms. The holograms consists of PSFs from point-sources placed at different positions and recorded in the Fresnel approximation using 2 parabolic wavefronts, Equation (6), on top and in the close-proximity near field using 1 spherical wave, Equation (3), below. In Figure 4a the WV PSR is shown to exhibit an interference pattern stronger in intensity than the signals themselves and located in between them. Self-interference in the vicinity of non-linear sections of non-linear signals can be observed well in Figure 4d by the broadening along the curved sections of the PSF.

Another common PSR is the spectrogram. As can be seen in the middle column of Figure 4, it does not introduce new false signals; however, it has lower resolution. Although its resolution can be increased by means of high-resolution STFTs [11], it will always remain inferior to the comparatively recent S-method [12] depicted in the right column of Figure 4. A similar argument was already made for the superior resolution of the smoothed pseudo Wigner–Ville representation over the spectrogram in [41]. The smoothed pseudo Wigner–Ville representation is a special case of the S-method.

In the following, we will introduce first the spectrogram and then the S-method, because the efficient computation of the more general S-method is based on the spectrogram. Section 3.4 and Section 3.5 thereafter will explain how to apply the PSRs to DHs and list the advantages and disadvantages of both methods essential for a visual interpretation of holograms.

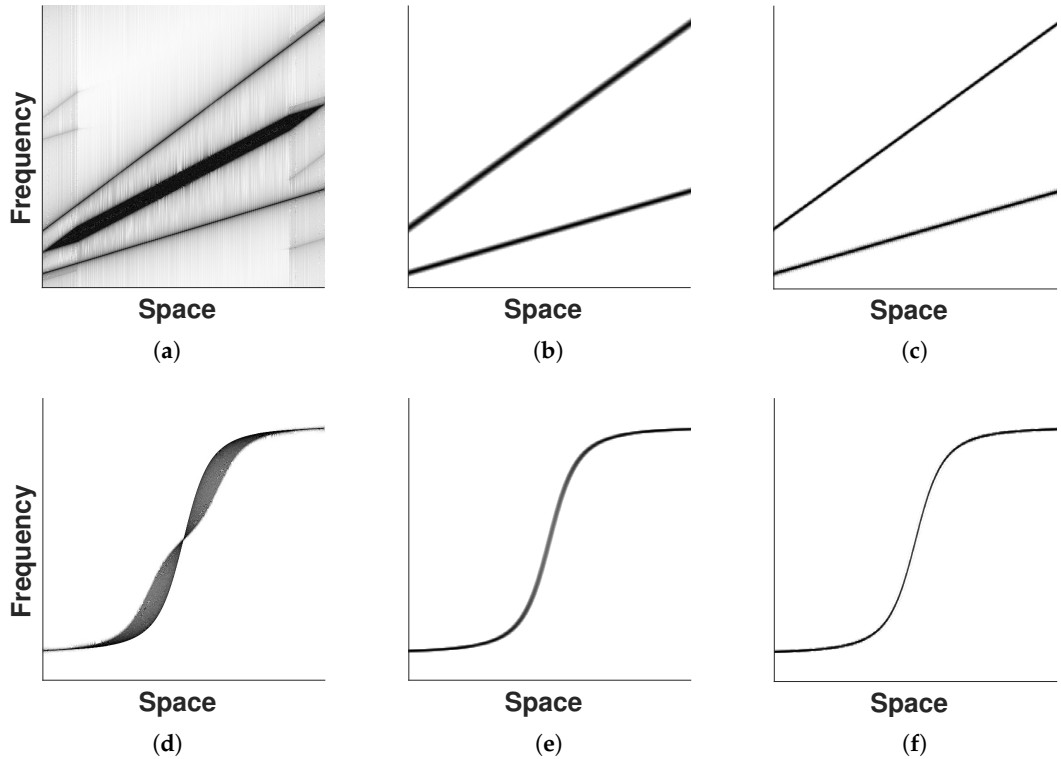

**Figure 4.** Comparison of phase space representations evaluated on two examples of wavefronts expanding along $z$ illustrate the arising interference terms—$2\times$ parabolic (top); $1\times$ spherical (bottom). (**a**) Wigner–Ville; (**b**) spectrogram; (**c**) S-method; (**d**) Wigner–Ville; (**e**) spectrogram; (**f**) S-method.

*3.2. Introduction to Spectrograms*

Let $g \in \mathbb{C}^N$ be a discrete 1D complex-valued signal of length $N$ and $w \in \mathbb{C}^M$ be another function, called (spatial) window $w$ with $M \leq N$, e.g., Figure 5a. We will refer to $M := 2M' + 1, M' \in \mathbb{N}_0$ as the window length. Contrary to analyzing the signal in its native (space) domain or its Fourier domain, we strive to understand when which frequency components are active. Due to the perfect delocalization of sinusoids, the elementary components of a Fourier transform, we can only do so by taking the Fourier transform of our signal over many small intervals. This is where the window function comes into play. The element-wise (Hadamard) product $g(\xi^{px} + m)w(m), \xi^{px} \in \{0, \ldots, N-1\}, f_\xi^{px} \in \{0, \ldots, M-1\}$ will have only at most $M$ non-zeros (in case $\xi^{px} + m$ exceeds $[0, N-1]$ the signal $g$ is continued periodically). A discrete Fourier transform of $gw$ will provide a frequency analysis localized on the at most $M$ coefficients of the signal selected by the window. This local frequency analysis will be quasi-stationary in that it assumes a periodic continuation of the signal beyond the $M$ non-zeros of the window and constant frequencies within, see Figure 5b. Shifting the window with respect to the signal and re-evaluating the Fourier transform each time, will provide a (quasi-stationary) representation of excited frequencies over space. Mathematically this concept of a windowed Fourier transform is known as the short-term Fourier transform (STFT), which is given at a location $\xi^{px} \in \{0, \ldots, N-1\}$ (around the window center) and a frequency $f_\xi^{px} \in \{0, \ldots, M-1\}$ (excited within this window) as

$$\mathrm{STFT}_w(g; \xi^{px}, f_\xi^{px}) := \sum_{m=0}^{M-1} g\left(m - M'\right) w\left(\xi^{px} + m - M'\right) e^{-2\pi i f_\xi^{px} m / M} \tag{10}$$

and the spectrogram $SP_w(g; \xi^{px}, f_\xi^{px})$ is then defined as

$$SP_w(g; \xi^{px}, f_\xi^{px}) := \left| \mathrm{STFT}_w(g; \xi^{px}, f_\xi^{px}) \right|^2. \tag{11}$$

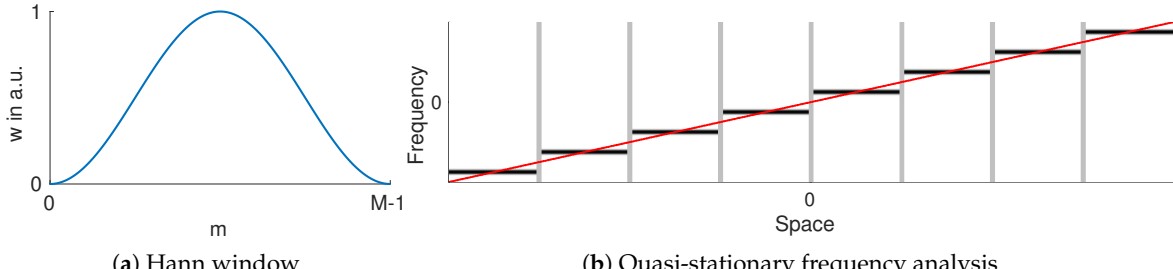

(**a**) Hann window           (**b**) Quasi-stationary frequency analysis

**Figure 5.** (**a**) Exemplary spatial short-term Fourier transform window. (**b**) Illustration of quasi-stationary frequency analysis of the signal (red). Within each spatial window (within gray verticals) frequencies are stationary. Here, the strongest response per window occurs at its average frequency.

While general window functions are possible, we shall content ourselves here with the simplest case of a symmetric, non-negative (real-valued), Hann window ([11], p. 532) of length $M \geq 1$, defined as

$$w(m) := \frac{1}{2}\left(1 - \cos\left(\frac{2\pi m}{N}\right)\right) = \sin\left(\frac{\pi m}{M}\right)^2, \quad m \in [0, M-1]. \tag{12}$$

and normalized in the $\ell_2$-norm via

$$\sum_{m=0}^{M-1} |w(m)|^2 = 1, \tag{13}$$

to preserve the $\ell_2$-energy of the signal.

A window of length $M$ allows for $M$ distinct frequencies to be analyzed. Thus if $M$ is decreased, the spatial resolution is increased (as more distinct windows exist) but the frequency resolution (per window) decreases. If the converse is true, the frequency resolution is increased and traded for spatial resolution. A formal result of this is Lieb's uncertainty principle for the short-term Fourier transform [39]. Selecting $M = \lfloor\sqrt{N}\rfloor$, where $\lfloor\cdot\rfloor$ is the nearest lower integer, will provide the most balanced space-frequency resolution. In practice, overlapping windows are used to enhance the resolution through an increase of the redundancy in the representation. Good results are achieved by choosing maximal overlap $M-1$ between long windows, e.g., setting $M = 4\lfloor\sqrt{N}\rfloor$ as was done in Figure 4. [42] describes how window size affects the confidence of instantaneous frequency estimates.

Despite these measures, the joint resolution of the spectrogram remains low. Further enhancements are possible by abolishing the invertibility of the underlying STFT. For narrow-band signals, which excite only a few frequencies at every location, high-resolution schemes can be found in [11,43]. However, as the considered DHs of diffusely scattering, macroscopic scenes are wide-band signals, another way of increasing the resolution is required and will be shown in the next section.

### 3.3. Introduction to the S-Method

Given Equation (10), the S-method is defined with the STFT and an arbitrarily chosen frequency filter window $v \in \mathbb{C}^L$ of length $L := 2L' + 1, L' \in \mathbb{N}_0$ as

$$SM_{v,w}(g; \xi^{px}, f_\xi^{px}) := \sum_{l=0}^{L'} [v(L'+1-l) + v(L'+1+l)] \quad \cdot$$
$$\overline{\mathrm{STFT}_w\left(g; \xi^{px}, f_\xi^{px} + L'+1-l\right)} \, \mathrm{STFT}_w\left(g; \xi^{px}, f_\xi^{px} + L'+1+l\right), \tag{14}$$

with complex conjugation $\overline{\phantom{-}}$. The length and functional form of the spatial window $w(m)$ can be chosen as described in the previous section. One good candidate for $v$ is a short Hann window. Its length determines the frequency resolution of the transform and thus should be small, e.g., $L = 5$. In addition, all other parameters influencing the quality of the STFT, will affect the visual performance of the

S-method. The most simplistic and efficient implementation for symmetric and real-valued frequency windows $v$ is explained in great detail in the original contribution [12] and in [11]. It is restated in Listing 1.

**Listing 1.** An efficient S-method implementation

---

*S-method* ( STFT $\in \mathbb{C}^{M \times N}$, frequency window $v(l) \in \mathbb{R}^L, l \in \{0, \ldots, L-1\}$ with peak at $L'+1$ )
　　$SM = |STFT|^2 \, v(L'+1)$　　　　　　　　　　　　　　　　　　　▷0.th term is the spectrogram
　　**For** ( $l \in \{0, \ldots, L'\}$ )　　　　　　　　　　　　　　　　　　　　　　▷l.th refinement
　　　**For** ( $f_\xi^{px} \in \{l, \ldots, M-1-l\}$ )
　　　　**For** ( $\xi^{px} \in \{0, \ldots, N-1\}$ )
　　　　　$u = \mathcal{R}eal(STFT(f_\xi^{px} + L' + 1 - l, \xi^{px}) \overline{STFT(f_\xi^{px} + L' + 1 + l, \xi^{px})})$
　　　　　$SM(f_\xi^{px}, \xi^{px}) = SM(f_\xi^{px}, \xi^{px}) + [v(L'+1-l) + v(L'+1+l)]u$
　　**return** $SM \in \mathbb{R}^{M \times N}$

---

As with the spectrogram, there exist also extensions to the S-method [11,44], which can improve its resolution further, but they are not considered here for the sake of clarity. Instead we note, that several other PSRs are contained in the S-method (not to be confused with the S-transform) as special cases.

1. If $v(0) = \delta(0), L' = 0$ one obtains the spectrogram, that is $SM_{v,w} = SP_w$. Thereby $w(m)$ is commonly normalized by Equation (13).
2. If $v(0) = 1, L' = 0$ one obtains the pseudo Wigner–Ville representation [40], which initially was constructed as restriction by windowing of the space-lags $w(m)$ considered during the computation of the Wigner–Ville representation. It is $SM_{v,w} = PWV_w$.
3. If $v(0) = 1, L' = 0$, and $w(0) = 1, M' = 0$ one obtains the regular Wigner–Ville representation, that is $SM_{v,w} = WV$.
4. If $v(l), w(m)$ are of even parity, and the following standard requirements hold $\mathcal{F}(v)(\zeta = 0) = 1$, $v(l) \in \mathbb{R}, w(m) \in \mathbb{R}$, one obtains the smoothed pseudo Wigner–Ville representation [40], that is $SM_{v,w} = SPWV_{v,w}$. $\mathcal{F}$ signifies the discrete Fourier transform.

*3.4. On the Application of Phase Space Representations to Digital Holograms*

Rather than studying the entire 4D phase space of DHs, we will consider only 1D cross-sections (see Figure 1b) in the form of individual rows or columns sampled from the DH. Those provide, sufficient information for most analysis tasks. For near-field holograms, the row/column positions correspond to horizontal or vertical slices in scene space and should be chosen according to the objects of interest. For far-field holograms, they correspond to frequency sub-bands and thus reveal information about the entire scene at once. Unless stated otherwise, we will evaluate the PSRs on the middle row of the DHs, for example given a hologram $H \in \mathbb{C}^{N_\eta \times N_\xi}$ as $SM_{v,w}(H(:, \lfloor N_\eta/2 \rfloor); \xi^{px}, f_\xi^{px})$.

*3.5. Important Properties for Phase Space Visualization of Digital Holograms*

The spectrogram and the S-method are both visually faithful representations in that they share two mathematical properties which are essential for an accurate visual interpretation. First, as members of Cohen's class they transform linear under space, frequency shifts of the signal and, second, they largely avoid introducing artificial signal components. Thus, if a signal component is visible in either PSR, it was actually present in the signal at these phase space coordinates.

The first property is captured by the notion of covariance under space and frequency translations. That is, if a signal $g$ is shifted in space or frequency, the PSR $\rho$ shifts by the same amount:

$$\forall \Delta\xi, \Delta f_\xi \in \mathbb{R}: \quad h(\xi) = g(\xi - \Delta\xi)e^{2\pi i \Delta f_\xi} \rightarrow$$
$$\rho(h; \xi, f_\xi) = \rho(g; \xi - \Delta\xi, f_\xi - \Delta f_\xi). \tag{15}$$

Covariance is essential to locate a holographic signal in phase space, see for example Figure 2. For near-field, non-Fourier hologram lateral translations of the object are reflected by lateral translations in the phase space of the object wave. Frequency shifts of the recorded hologram occur, for example when tilting a plane wave illumination. We will show explicit examples in Section 4 and Section 5.5.

Artificial signal components introduced through "cross-talk" of multiple signal components in the original signal are a mathematical necessity and their position and shape has been thoroughly studied in the past. It is understood for the entire Cohen class [40], and can render PSRs useless for the visual interpretation of many component signals such as DHs. Thus, it is worth understanding how they are handled in the spectrogram and the S-method, as to still allow for a clear visual interpretation.

The spectrogram hides cross terms by locating the "cross-talk" on top of the actual signal. One speaks of inner interference terms in this case. Inner interferences lead to a broadening of the signal in phase space, perceivable as low resolution.

The S-method follows a different strategy. Implicitly it introduces outer interferences, which are located next to the original signal, but they are filtered out via the frequency filter window. This guarantees a high-definition of the signal components itself, which is comparable to that of the WF PSR. We quote [11], p. 647: "The S-method can produce a representation of a multi-component signal such that the distribution of each component is its [pseudo] Wigner distribution, avoiding cross-terms, if the STFTs of the components do not overlap in time-frequency plane." (Note: time-frequency plane = phase space). As the frequency filtering is imperfect, weak cross-terms can occur when signals overlap in phase space or are in very close proximity. In either case they do not impair visual interpretation and the visual degradation is trumped by far by the increased phase space resolution compared to the spectrogram.

In Figure 6 both PSRs were computed with equal parameters ($N = 8192$, $M = 1024$, overlap length $M - 1$, $L = 5$) and the spatial separation of the dices is better visible with the S-method. Note, because the Fourier transform of a Hann window is given as a sum of 3 delta distributions, for $M > 3$ its frequency resolution is better than its spatial resolution. Thus, a long spatial Hann window will act as a stronger frequency filter than a spatial filter. This is why the inner interferences in the spectrogram appear to be smeared out in the spatial domain in Figure 6a. Conversely, the additional Hann frequency filter window of the S-method, acts as a stronger spatial than frequency filter and recovers the spatial phase space resolution.

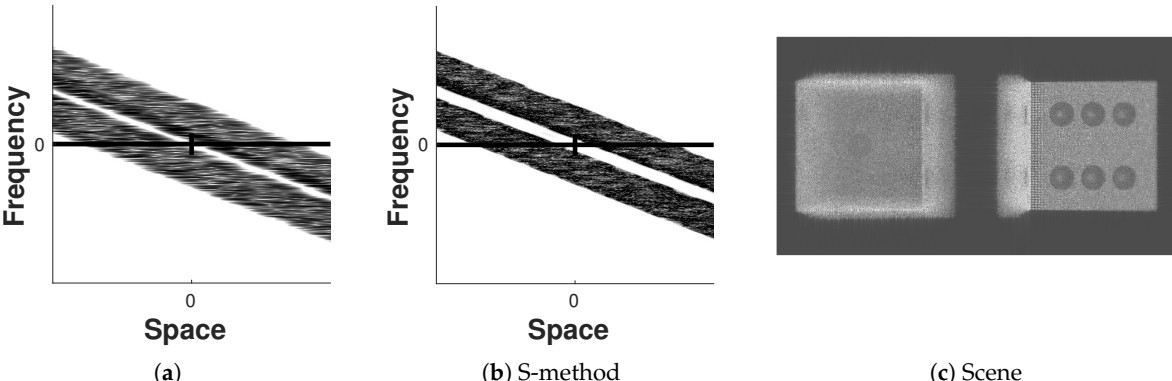

(**a**) (**b**) S-method (**c**) Scene

**Figure 6.** A hologram of 2 dices in close proximity is used to contrast the phase space resolution of the spectrogram and the S-method. A back-propagation of the scene is shown in (**c**). (**a**) Spectrogram; (**b**) S-method; (**c**) scene.

Both the spectrogram and the S-method can be computed much faster than the WV PSR, due to the reliance on small filter windows.

#### 4. Capture of Holograms: Interference of Object and Reference Waves

Interference is an essential concept used in physical implementations of holography. It enables to record off-axis holograms [45] where the image of the light source, the conjugated, and the real image are laterally separated in space upon back-propagation of the hologram. While the optical wavefront may only be partially characterized in specific setups, solely four types of reference waves, denoted in Equation (1) as $E_R$, are common.

**Diverging spherical wave** This is the wavefront of an ideally isotropic (reference) point-source. It is used for rigorous propagation within the confines of scalar diffraction theory. An exemplary phase space footprint is shown in Figure 3c.

**Parabolic wave** This is an approximation to a spherical wave in the Fresnel or paraxial approximation. It is used for Fresnel propagation. An exemplary phase space footprint is shown in Figure 3c.

**Planar wave** (also: plane wave) These form a dual basis to spherical waves, in the sense that they can be interpreted as spherical waves with infinite curvature. This wavefront is used for accurate propagation with the angular spectrum method. In phase space the footprint will be line parallel to the spatial axis. A planar wave incident with an angle $\theta \neq 0°$ will appear as a line at frequency $f$ given by Equation (5).

**Other** Other forms can arise e.g., in shearing interferometry when the object wave is used to interfere with itself. The phase space footprint will depend on the object.

In case the individual object and reference wavefronts are known, one can study the interference in phase space with PSRs—where it is understood intuitively as a sum of the corresponding instantaneous frequencies. We showcase the phase space footprint of exemplary holograms obtained from the interference of an object and multiple reference waves in Figure 7. As object wave we use the scene of Figure 6, recorded in the paraxial limit, after complex conjugation and up-sampling it by a factor of 4. Whilst the former modification merely aids visualization, the second avoids the occurrence of aliasing after interference.

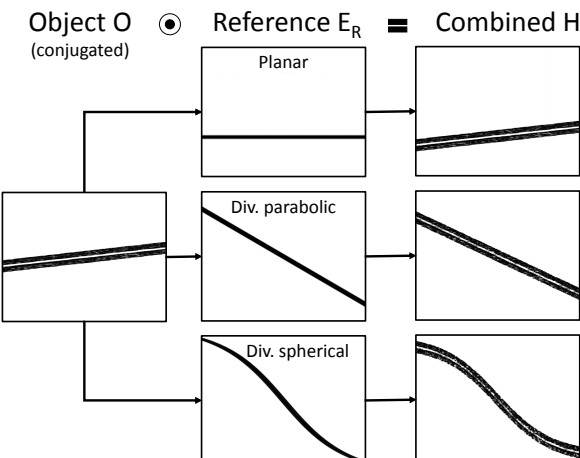

**Figure 7.** Geometric interpretation in phase space of the Hadamard product of an object wave *O* and distinct reference waves $E_R$, see Equation (1), yielding a hologram. *O* used in this figure is the complex-conjugated wave of Figure 6, after up-sampling it by a factor of 4 in order to avoid aliasing.

#### 5. Applications of Phase Space Representations in Digital Holography

##### 5.1. Exploring an Arbitrary Hologram

Now, let us consider some typical questions posed for a hologram. Imagine being presented with a hologram, in its complex-valued representation, without any additional information aside from pixel pitch and its wavelength.

**1. What is the type of hologram?** If a single recording wavelength has been used, the hologram is monochrome and the reasoning from Section 2 may be used as follows: for holograms of macroscopic objects, recorded at some out-of-focus plane, the amplitudes in the spatial and in the frequency domain (shown as projections in Figure 8c) usually do not reveal anything about the content of the 3D scene.

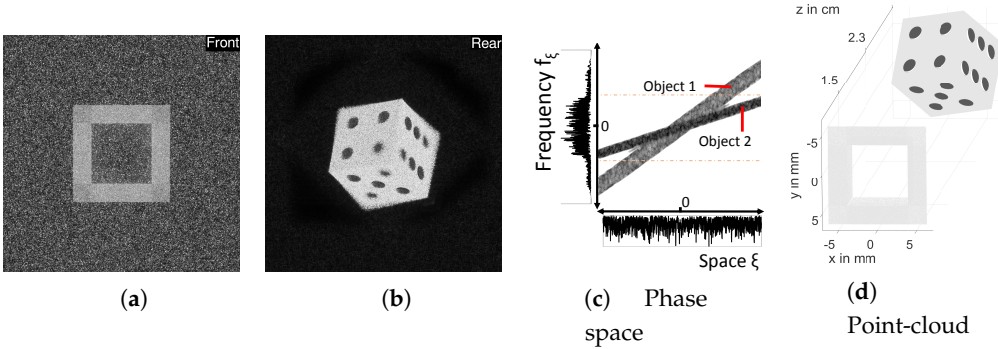

**Figure 8.** (**a**,**b**) Front and rear back-propagation of a hologram containing two objects. Only by joint consideration the entire scene is visible. (**c**) The S-method evaluated on a 1D cross-section of the hologram, shows a clear separation of (all) objects in phase space. Objects are indistinguishable in either spatial or frequency domain alone. (**d**) For reference, the point-cloud model of the scene is given. Parts of this figure are reused from [30].

Note, the intensity variation in frequency domain does not exclusively correspond to the individual objects. However, the S-method calculated from the central 1D cross-section immediately aids understanding of the scene composition. The phase space footprint is composed of parallelograms, suggesting that Fresnel propagation is sufficient for back-propagation, compare with Figure 3c.

**2. What is depicted in the hologram?** If the hologram is of sufficiently large resolution and shows in the S-method wide parallelograms instead of thin lines, compared with Figure 4c, then it depicts dense sets of point-sources in close proximity. This can be understood by reinterpreting the object wavefield $O$ in Equation (1) as a spatial convolution of the amplitude-phase distribution $h(x, y, z)$ of point-sources in the scene with a slightly varying PSF located at $(x, y, z > 0)$, corresponding to a single line such as in Figure 3c. We can thus identify 2 objects in the present hologram. Using Equation (7), we see that the smaller slope of object 2 corresponds to a larger depth. For reference, we provided a point-cloud model of the scene in Figure 8d. It consists of a partially occluding spyhole in front of a dice. Using this as reference, we identify the rear object, object 2, as the dice.

*5.2. Optimizing Space-Frequency Bandwidth Product*

The part of the signal in phase space, that is actually conveyed by a pixel-matrix depends exclusively on its pixel pitch $p$ and number of pixels $N_\xi \times N_\eta$. Due to physical limitations on the pixel pitch and addressable number of pixels, the phase space of visible light holograms is never captured in full. In case of in-line or on-axis holography, these limitations of any row/column of said matrix defines a rectangular, axis-aligned bounding box around the optical axis $(\xi, \eta) = 0$ in the phase space. If we assume effectively a complex-valued modulation/detection capabilities per pixel, the bounding box is given by the pixel matrix—bandwidth $1/p$ times the spatial extent of the hologram given by $Np$ in absolute units of frequency [Hz] and space [m]. This holds for digital and optical setups [36] alike, if aberrations and intermediate limiting numerical apertures of optical setups are neglected. The space-bandwidth product (SBP) of a spatial light modulator (SLM) is given as $\text{SPB}_{\text{SLM}} := N_\xi N_\eta$ and cannot be enlarged with additional optics. It characterizes a passive optical system through its entendu (active area times angular spread) [36].

We illustrate the bounding box using normalized coordinates in Figure 9c. For this we selected a hologram depicting a single model of a diffusive scattering "Earth" (from the INTERFERE-II database [46], see Figure 9a) with a resolution of 8192 × 8192, 1 μm pixel pitch, and a wavelength of 633 nm. The model is located at a depth of 1.18–1.64 cm and therefore comparatively shallow.

The ramifications of this bounding box interpretation are substantial. It allows to predict and subsequently tune systems towards more effective use of the SBP by maximizing the ratio

$$\frac{\text{SPB}_{\text{H}}}{\text{SPB}_{\text{SLM}}}, \tag{16}$$

where $\text{SPB}_{\text{H}}$ is the SPB of a signal *H*—see [13], p. 33f. Conceptually $\text{SPB}_{\text{H}}$ equals the area occupied by the signal within the bounding box and we will discuss two ways of obtaining a numerical estimator in Appendix A. For example, a holographic capture of only object 2 of Figure 8 requires about half the frequency bandwidth (i.e., only a hologram of double the pixel-pitch and half the number of pixels) than a recording of object 1.

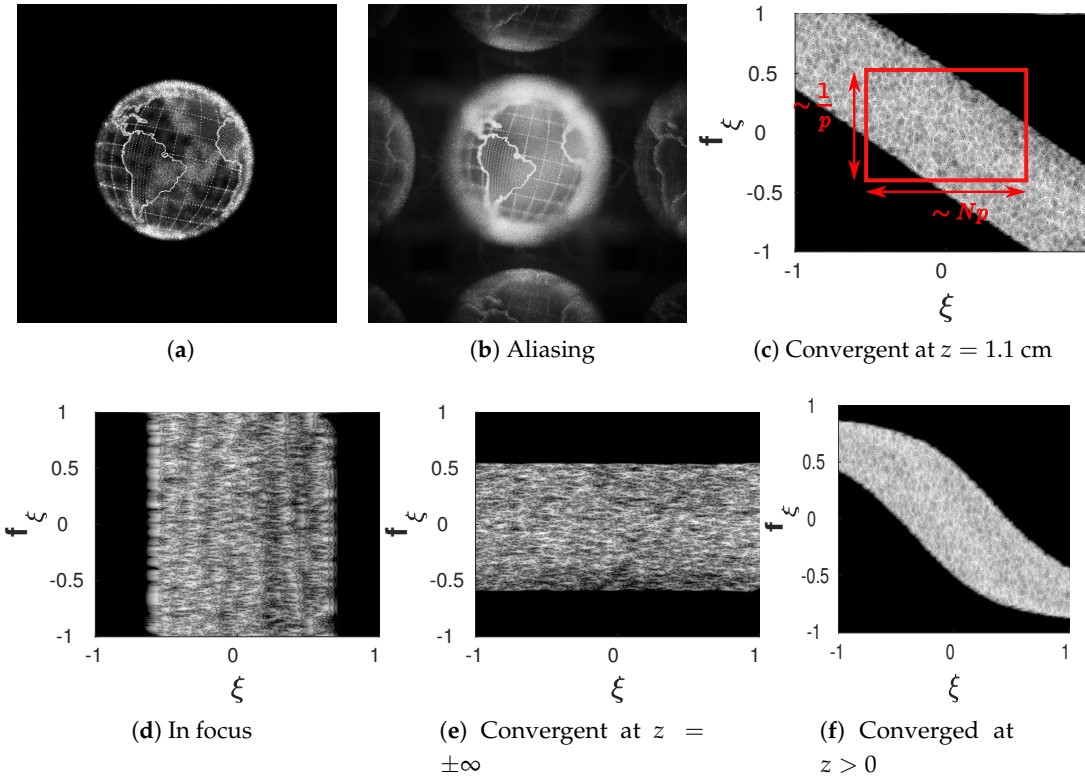

**Figure 9.** (**a**) Standard back-propagation of a hologram of a diffuse Earth recorded in the Fresnel limit using a numerical aperture of 1/4 of its size. (**b**) Back-propagation after undersampling the hologram by a factor of 1/0.65 without using a numerical aperture. Phase space footprints of 1D cross-sections are shown after refocusing to different distances in (**c**–**e**). In (**c**) the space-frequency bandwidth boundary box is marked. (**f**) Shows a hologram of the same object in a different diffraction regime.

All signal components that exceed the maximal supported frequency of a DH $1/(2p)$ (also: Nyquist frequency) cause aliasing. Aliased frequency components appear as being wrapped around said frequency range. During the reconstruction of weakly aliased holograms, parts of the scene appear mirrored along the boundaries. See for example Figure 9b where we reconstructed the hologram (without any aperture and increased contrast) after resampling the hologram with a pixel pitch of $1/0.65$ μm. As the hologram is now undersampled, aliasing artifacts arise along the borders. For more more severe undersampling, the aliased images and the original will overlap and greatly reduce visual quality.

In optical display setups aliasing will be visible in the form of higher diffraction orders. Two contributing factors are to be distinguished, see Figure 5 in [36]. One hand the pixelated structure of the SLM can give rise to "SLM orders" in the far-field. On the other hand the displayed content gives rise to "CGH orders" due to undersampling or due to imperfect modulation. In the on-axis case, the CGH orders are located at lateral extents larger than $(\lambda z)/p$ after propagating the distance $z$ [47].

The maximal frequency occupied by the hologram within its SBP, is proportional to its transversal spatial resolution in scene space for small to medium numerical apertures. The resolution (defined by the Rayleigh criterion [13]) after propagation of $z$ can be calculated within the paraxial approximation [47] via Equation (7) as

$$\text{Res} = \frac{Np}{\lambda z} \,. \tag{17}$$

### 5.3. Depth Estimation

The discussions of Sections 2 and 4 can be combined to provide a rough scene depth estimate for holograms where such information is not accessible, as we indicated in Section 5.1. For microscopic scenes with only a few scattering centers even accurate 3D positioning is possible [16,25].

As a starting point, we note that the combined wavefields in Figure 7 are in general of unique shape and through analysis it is possible to estimate their focal distances. Figure 9 demonstrates how the phase space footprints shape varies for the hologram of a comparatively shallow object ($N_{Fr} \sim 1650 - 2250$).

In case a signal shows as impulses (lines parallel to frequency axis), e.g., Figure 9d, the hologram is in focus, and all point-sources located within the depth of focus are already converged to points (Dirac impulses in the 1D cross section). The impulses will be bandwidth-limited by the holograms phase space bounding box. The lateral resolution is limited by the pixel pitch in the object plane and potentially by diffraction—see the Rayleigh criterion in the former section, in the case of the paraxial approximation. These holograms have been recorded as image plane holograms or have been refocused prior to their analysis. Note, that Figure 9d shows the phase space footprint associated with a horizontal center cut through Figure 9a. Curiously, in Figure 9d it is possible to make out individual PSF clusters with the S-method that remain slightly out of focus, due to the sparsity of the point cloud model. Their slope informs about whether they are located in front or behind the current focal plane. This proves helpful in identifying the ordering of multiple sparse objects in depth [25]. A situation which appears frequently in digital holographic microscopy, where usually sparse, multi-component signals are being analyzed. Traditional methods create a depth map relying on exhaustive back-propagations to different scene depths.

In case a signal extends over entire spatial domain and appears to be made up of horizontal lines, e.g., Figure 9e, the wavefield does not converge at any finite distance. Such a hologram has been recorded as a Fraunhofer or a (lensless) Fourier hologram. In essence, a Fourier transform is required for refocusing. This representation is especially advantageous in experimental setup as it makes optimal use of the space-frequency bandwidth product $\text{SPB}_H$ and was proposed for use in [32] under the name "compact space-bandwidth representation".

Whenever a parallelogram with angles sufficiently different from $90°$ and $0°$ shows, e.g., Figure 9, the focal plane is at some distance $z > 0$ or $z < 0$ depending on the sign of its slope. These holograms have been recorded as Fresnel holograms in the paraxial limit or in the near field with a small frequency bandwidth due to a large pixel pitch. Depending on which, at least the Fresnel method or the rigorous back-propagation need to be used.

Finally, if a sigmoidal shape as in Figure 9f is seen, the focal plane is at some distance $z > 0$ and the rigorous method needs to be used for refocusing—usually the angular spectrum method is employed.

For finite propagation distances, i.e., in the latter two cases, $z$ can be estimated from a parametric fit of the central ridge of the footprint or from the footprint of an isolated, bright scatterer. Without aberrations a sigmoidal (Equation (4)) or linear curve (Equation (7)) are to be used. We provide an explicit example in Appendix B. Depth estimation through curve fitting is also the essence of the

filter matching technique used in online synthetic aperture radar processing or used to estimate the distances of a scattering target from the radar site in inverse synthetic aperture radar imaging for motion compensation, see [7] p. 805ff. Instead of curve fitting in phase space, one can also estimate the optical flow, computed from the STFT, to guide depth estimation as done in [48] or use more elaborate estimation schemes of the instantaneous frequency dependence of the brightest scene parts through interleaved confidence intervals and adaptive window sizes as detailed in [42].

### 5.4. Diagnosis of Distortions of Digital Holograms

In this section, we will demonstrate how visually interpretable PSRs can be used to reliably determine certain sources of error in a DH that were either introduced during processing or generation.

### 5.4.1. Quantization Errors

Aside from the aliasing artifacts discussed in Section 5.2, PSRs can also be used to analyze flaws that occurred upon processing. Probably the most frequent distortion encountered by DH is quantization. May it be for reasons of limited analog-digital converter precision in the recording sensors, limited bit-depth in spatial light modulators, or reasons of a limited storage capacity, quantization is frequently applied. Multiple works have been published on the effects of quantization in DH, e.g., [49–51], and in general it must be said that the perceptual quality impairment of quantization depends strongly on the hologram type, its diffraction regime, the pixel pitch, the specific holographic display setup, and especially the distance of the scene to the hologram plane. Thus for each setup type and set of hologram parameters a perceptual quality evaluation is currently required.

One independent way to assess the severity of the effects of quantization on the hologram is by visualizing its phase space footprints as done in Figure 10 for the object of Figure 9 using the half the dynamic range to enhance contrast. Multiple bit-depths were used with a uniform quantizer applied to real and imaginary parts independently. Clearly, we can see the increase of quantization noise, which is typically modeled as white noise per pixel. White noise affects all frequencies and since it is delocalized, we expect a drastic worsening of the visible signal to noise ratio, such as in Figure 10. At the chosen contrast almost no artifacts are visible with 5*bit* per channel. While the analysis of quantization artifacts is important, PSRs can be used to analyze arbitrarily complicated post-processing chains of DH.

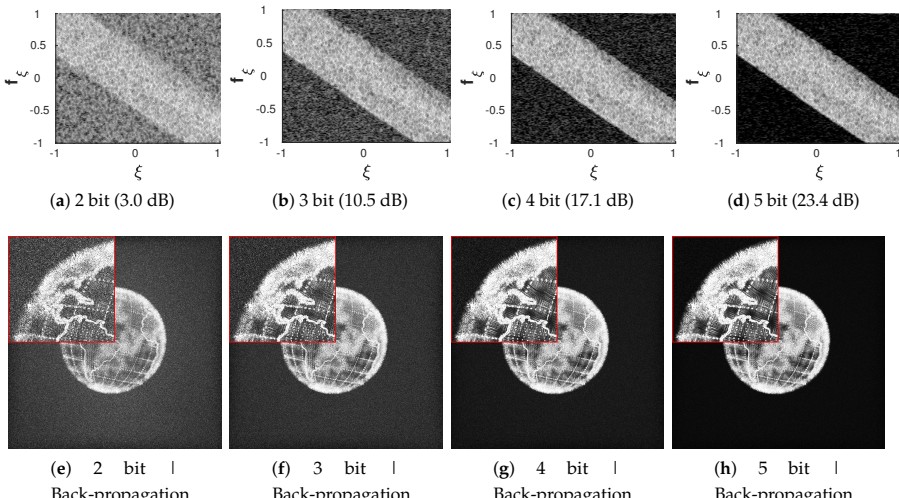

(**a**) 2 bit (3.0 dB) (**b**) 3 bit (10.5 dB) (**c**) 4 bit (17.1 dB) (**d**) 5 bit (23.4 dB)

(**e**) 2 bit | Back-propagation (**f**) 3 bit | Back-propagation (**g**) 4 bit | Back-propagation (**h**) 5 bit | Back-propagation

**Figure 10.** Top: Phase space visualization of the error introduced in Figure 9c by uniform quantization with various bit-depths per real/imaginary part. In brackets the signal-to-noise ratios of the quantized holograms are provided. Bottom: Corresponding back-propagations using the same parameters as in Figure 9a. The dynamic range was halved to enhance contrast.

### 5.4.2. Tracing Missing Space-Frequency Information

There might be various reasons, for why specific space-frequency information is missing from a DH. In Section 5.2, we explained two of the most common reasons. Whenever the phase space footprint of a signal is clipped along the spatial dimension, the wavefront expanded too far and exceeds the (virtual) detector. If the footprint is clipped in the frequency dimension the signal is undersampled. While those cases are easy enough to recognize for holograms presented in their native hologram plane (where they were recorded/generated initially in), it becomes challenging when one or multiple post-processing steps are applied. An example is provided in Figure 11a where bandwidth-limitation is visible on the two chirp bundles belonging to objects in the front after propagation to the scene middle. Other examples can be a result of motion compensation, see e.g., Figure 12b,e, or other manipulations.

The missing information may impair the lateral scene resolution only slightly, see Equation (17), and will be thus hard to notice. A glance with a PSR will, however, quickly reveal such flaws and allow optimization for optimal performance.

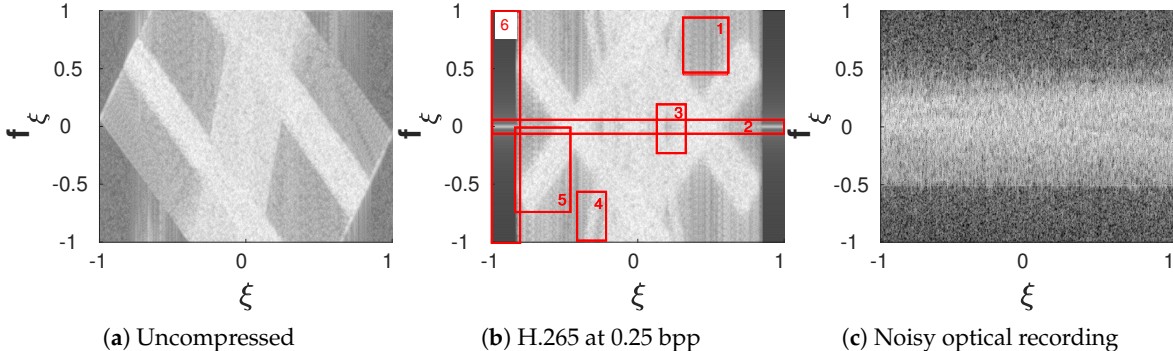

(**a**) Uncompressed　　　　　　　(**b**) H.265 at 0.25 bpp　　　　　　　(**c**) Noisy optical recording

**Figure 11.** (**a**,**b**) Phase space footprint of 1D cross-section of un-/compressed (with H.265/High Efficiency Video Coding (HEVC)) multi-object hologram "DeepChess" from Interfere-IV [46]. (**c**) Noise in the optical holographic recording "Squirrel" from Interfere-IV [46] of an object with metallic surface.

### 5.4.3. Compression of Static Holograms

Another "distortion" one might encounter when working with DHs is lossy compression. To minimize coding artifacts and maximize compression efficiency the compression scheme needs to be tailored to the typical phase space footprints of the signals under consideration. In transform-based coding, typically a best tiling strategy of phase space is determined for a given class of signals and a suitable frame or basis is selected based on it. PSRs are a vital instrument in this design process nowadays, and have been in the past. Once such an optimal tiling has been found and a basis representation has been derived, the latter can subsequently be used to reduce the signals to a few essential coefficients and subsequently efficiently compress signals. Whilst the initial motivation for new basis designs does not necessarily have to be sourced in phase space analysis, for some signals, especially for DH, this approach is highly advisable and in line with advances in mathematics made of the past few decades such as: STFT, wavelet, or Gabor schemes. An abundance of work followed the introduction of each of these schemes, with applications to all kinds of one- or two-dimensional signals. See e.g., [52] or more specific to DH [17,28,53]. In addition to basis design, PSRs can also be used to simply, localize coding artifacts and identify discarded information for entire coding strategies by comparison of the PSRs of un-/compressed holograms. See for example Figure 11b, where we present the "DeepChess" hologram from the INTERFERE-IV database [46] compressed with H.265/High Efficiency Video Coding (HEVC) in intra-mode at a low bitrate of 0.25 bit per complex sample. The codec was chosen as an example here since it is one of the two anchor codecs proposed for the JPEG Pleno on holography. The codec is applied after a uniform 8 bit quantization of the entire dynamic

range of the real and imaginary channels of the hologram. The highlighted differences can be identified and explained as follows:

1　reduced signal to noise ratio due to the introduction of discontinuities (impulses) through block based coding, and use of a basis, poorly localized in frequency (discrete cosine transform with short spatial windows $w$, cf. Section 3.2).

2+5　added DC term and opposite diffraction orders, due to separate compression of real and imaginary parts, cf. Section 5.5.5.

3+4　missing information due to the bitrate-distortion optimization being tuned for natural images.

6　clipped signal, due to thresholding during rate-distortion optimization.

Another important characteristic for the compression of especially optically recorded holograms is the amount of measurement noise. The more noise is contained in the signal, the larger its footprint in phase space and the more difficult it is to employ any sparsifying transform. An example of noise visible in phase space is shown on a 100 dB scale in Figure 11c for the optically recorded "Squirrel" hologram from INTERFERE-IV [46]. The high amounts of measurement noise in this case are a result of the metallic surface of the recorded object. Note, optically recorded incoherent measurement noise is similar to quantizing real and imaginary parts to low bitrates, see Section 5.4.1.

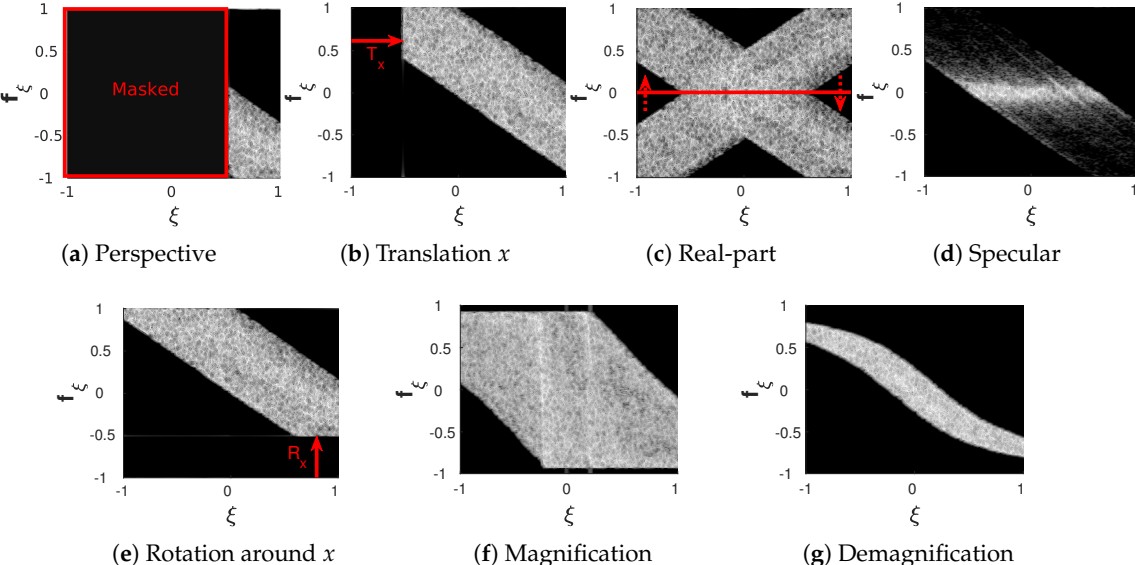

(**a**) Perspective　　　(**b**) Translation $x$　　　(**c**) Real-part　　　(**d**) Specular

(**e**) Rotation around $x$　　　(**f**) Magnification　　　(**g**) Demagnification

**Figure 12.** Phase space visualizations of 1D cross-sections of the hologram displayed in Figure 9 after undergoing various manipulations discussed in separate sections below. (**d**) shows the "Earth8KS" from INTERFERE-II database [46], which scatters specularly instead of diffuse. (**f,g**) are transversal de-/magnifications of the object shown in Figure 9f.

## 5.5. Some Simple, Global Operations on Digital Holograms

In this section, we show the effects of simple manipulations such as perspective reconstruction, rotations, or translations of the hologram. All the operations presented within this and the previous section may be combined to understand more complicated DH processing chains.

### 5.5.1. Perspective

A perspective back-propagation of an out-of-focus hologram (i.e., a non image-plane hologram) can be obtained in the near field by using a "pinhole" in the form of a sub-hologram sized aperture, which is applied to the hologram (matrix) $H(\eta^{px}, \xi^{px}) \in \mathbb{C}^{N_\eta \times N_\xi}$ in the hologram plane prior to the

propagation to scene space. For example, a square aperture $W$ of 25% the size of the hologram retaining the center right is applied to the hologram as

$$\widetilde{H}(\eta^{px}, \xi^{px}) = W \odot H \quad \text{with} \quad W(\eta^{px}, \xi^{px}) \; := \begin{cases} 1 & \text{, if } \eta^{px} \in [\frac{1}{4}N_{\eta}, \frac{3}{4}N_{\eta}], \; \xi^{px} \in [\frac{3}{4}N_{\xi}, N_{\xi}] \\ 0 & \text{otherwise} \end{cases}, \quad (18)$$

with the Hadamard product $\odot$. In an optical setup the aperture can be our eye-pupil or the aperture of a camera. The observers position relative to the hologram plane is given by the location at which the aperture is applied to the hologram. If the aperture retains the center right of the hologram as in the example above, a center right view is reproduced upon back-propagation of the hologram.

PSRs can provide interesting additional insights on the visible contents. Taking for example the defocused hologram of Figure 9 and retaining the center right via Equation (18), discards any information outside the aperture as shown in Figure 12a. As we can see all remaining information will be located near $f_{\xi} \approx -1$ in this case. That is the visual explanation of the common statement in DH literature, that large frequencies predominantly contribute to corner/side view (see also the grating equation Equation (5)) and therefore any distortions of high spatial-frequencies will impact predominantly those views first. Obviously this statement is not precise, in case of the near uniform distributions of spatial frequencies of Fourier-, Fraunhofer, and image-plane holograms, cf. Figure 9.

### 5.5.2. Rotation

Rotations of the scene around the optical axis, $z$, are directly mapped onto rotations of the hologram around $z$. Such rotations are observable in PSRs as changes of the signal envelope only when the lateral extents of the scene are not isotropic. Small rotations of the scene around the $x$ or $y$-axis, can be approximated in the paraxial approximation ($sin(\alpha) \approx \alpha$) by small tilts in the illumination, which in turn correspond to shifts along the frequency axes [54]. That is for the example of a rotation by $\alpha_x$ radians around $x$

$$\widetilde{H}(\eta^{px}, \xi^{px}) \approx H(\eta^{px}, \xi^{px}) \odot e^{\frac{-2\pi i}{\lambda}(p\eta^{px}\alpha_x)}. \quad (19)$$

This will be directly visible with PSRs that are covariant with frequency, cf. Figure 12e. The exact modifications due to rotations [29,55] around $x$ and $y$ axes for large angles are more difficult to interpret in phase space, as they involve resampling in Fourier space.

### 5.5.3. Translation

Translations along $x, y$ are mapped directly onto translations of the hologram along $\xi, \eta$, too. For example, a translation about $\delta_x$ px and $\delta_y$ px will lead to

$$\widetilde{H}(\eta^{px}, \xi^{px}) = H(\eta^{px} + \delta_y, \xi^{px} + \delta_x). \quad (20)$$

Sub-pixel translations can be achieved by multiplications with phase kernels in Fourier domain instead [54]. If the chosen PSR is covariant in space, see Equation (15), the PSR will shift along $\xi$ as the scene shifts in scene space along $x$, see Figure 12b. Translations along the optical axis, $z$, corresponds to wavefield propagation as covered in earlier sections.

### 5.5.4. Transition between On- and Off-Axis Holograms

Any on-axis hologram can be converted into an off-axis hologram by a few generic steps:

1. Fourier transform on-axis hologram.
2. Zeropad hologram to twice its length in every dimension.
3. Shift the padded hologram by half its original size in every dimension.
4. Inverse Fourier transform the hologram.

5. The real part of these manipulations is the off-axis hologram.

Steps 1–4 will produce an off-axis hologram containing only the +1 or −1 diffraction order depending on the direction of the shift in step 3. The hologram has a frequency range that is entirely positive or negative. Step 5 then converts the complex-valued hologram into a real-valued off-axis hologram, containing both of the first diffraction orders as is easily verified by looking at the footprint in Fourier domain. To reverse the conversion above, step 5 can be neglected, the direction of the shift in step 3 is reversed, and step 2 changes into a crop to the original size.

The intuition behind the steps is as follows: With steps 1,2,4, the pixel pitch of the hologram is halved, in case the hologram was band-limited. Steps 1,3,4 describe the interference with a plane wave angled at ±45°, see Section 4. Step 5 adds the conjugated order to the hologram, thereby making it real-valued. This is easily verified by realizing that the reflection in phase space converts for example up- into down-chirps and vice versa.

In the language of complex analysis, steps 1–4, describe the conversion of an arbitrary uni-modular complex phase signal (thus analytic signal) into a real-valued phase signal, which is possible for any bandwidth-limited uni-modular complex valued signal. Step 5 reflects the fact that, the imaginary part of any 1D analytic signal can be exactly computed from its real part by means of the Hilbert transform [40], p. 40 et sequentes. Although the Hilbert transform is not unique anymore in higher dimensions [56], the solution presented above, which is called single-orthant 2D Hilbert transform and based in the continuous case on much more profound mathematics [57], works sufficiently well in practice for interferometric imaging methods, e.g., [58].

PSRs help to visualize and interpret the process above and allow us for example to interpret the source of some artifacts after compressing real/imag parts independently, see Sections 5.4.3 and 5.5.5.

### 5.5.5. Splitting into Real/Imaginary Parts

One common representation of on-axis DH is the complex-valued matrix form, which is regularly split into real and imaginary parts for the purpose of, for example, compression [28,59]. As a result that most PSRs [7] are insensitive to absolute phases, such as the relative phase between real and imaginary part when evaluated separately, their PSRs will look similar. Compared to the footprints of the complex valued signal, the PSRs of the real signals will appear to be mirrored with respect to $f_\xi = 0$, $f_\eta = 0$ as the conjugated signal is added and the Fourier transform of any real-valued signal is symmetric along the frequency axis. Exemplary, the real part of an on-axis hologram is displayed in Figure 12c.

The mathematical reasoning alone, provides no intuitive interpretation of the change in shapes of the phase space footprints. In light of the former subsection, we recognize that decomposing a complex-valued on-axis hologram into real and imaginary parts means storing it as two off-axis holograms with a 90° phase shift ($\sin(90° + x) = \cos(x)$ and $e^{ix} = \cos(x) + i\sin(x)$).

In conclusion, this leads to the quite interesting but unfamiliar interpretation of real- and imaginary parts of an on-axis hologram as a mixture of the ±1 diffraction orders per part, while both parts are differ in phase. This is similar to a two-plane phase-shifted representation of a hologram [59].

### 5.5.6. Transversal Magnification

Another elementary operation on holograms of practical relevance is transversal magnification of holograms, to which there are two approaches. One can either choose to change the pixel pitch from $p$ to $mp$ with some real number $m > 0$ or keep the pixel pitch constant and resample the content.

In the paraxial limit, the pixel pitch can be changed by uniform rescaling [60]. The phase space footprint in a PSR will show unchanged as the maximal sampled frequency $1/(2p)$ becomes $m/(2pm) = 1/(2p)$ and thus stays constant when the spatial dimension is denoted in samples. In contrast, if the pixel pitch in the hologram is kept constant despite de-/magnification of the content, a common zoom operation in phase space will be facilitated by a resampling of the signal. The hologram of Figure 9f

was regenerated after de-/magnifying the scene with $m = \{0.5, 2\}$ in Figure 12g,f. This corresponds to the second case of resampling with a constant pixel pitch.

### 5.6. Analysis of Degree of Surface Roughness

PSRs can also be used to categorize recorded objects as specular or diffusive reflecting. While objects with a high surface roughness and diffusive reflectivity, scatter light in all possible directions and are thus also clearly visible under large viewing angles, specular objects are not. They are predominantly observable in the paraxial limit as a result of their smooth phase distributions along their surfaces. In computer-generated holography, surface roughness on the scale of $\pm\lambda/2$ is modeled through a uniform random phase offset of all (surface) point-sources in the scene. Adding random phases to, or increasing the surface roughness of, a specular object can thus be perceived as a transformation of said object into a diffusive reflecting object upon generation/recording. We demonstrate this with the Earth holograms taken from the INTERFERE II database [46]. The holograms generated from specular and diffusive reflecting point-clouds are shown in Figure 12d and Figure 9. Clearly, predominantly low frequencies are excited by the specular object, which according to the grating equation Equation (5), correspond to small diffraction angles.

### 5.7. Multi-Object Hologram Segmentation and Motion Compensation

In the spirit of spatial blind source separation, PSRs can also be used to segment a multi-object hologram into sub-holograms corresponding to individual objects, thereby segmenting 3D scene space. Whilst this approach is based on numerically invertible PSRs applied to the full 2D holograms instead of visually interpretable PSRs applied to 1D cross-sections, it was motivated by a good knowledge of the phase space footprints of DHs. Hologram segmentation and subsequent multi-object motion compensation for dynamic holograms with the Gabor PSR, whose individual contributions are well localized in phase space, was demonstrated in [30]. Herein, we reprint in Figure 13 the phase space footprint of the individual objects after segmentation of the multi-object hologram used in Figure 8. The power of such a technique is of course manifold, and to mention only two potential applications from dynamic DH: one can strive for accelerated computer-generation of holographic videos or for better holographic video codecs, predicting future from past frames.

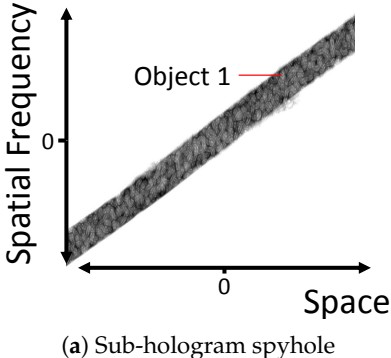
(**a**) Sub-hologram spyhole

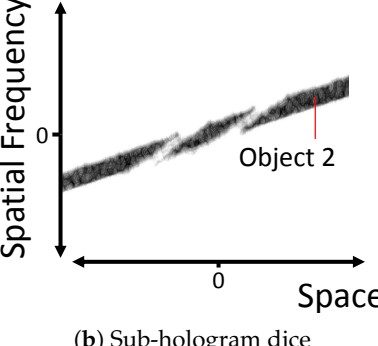
(**b**) Sub-hologram dice

**Figure 13.** (**a**,**b**) show the phase space footprints of the spyhole (Object 1) and the dice (Object 2) sub-holograms extracted with [30] from Figure 8c. Note, in (**b**) only the rims of the spyhole occlude the dice in this part of phase space.

### 5.8. Quality Assessment of Digital Holograms

A challenging task for DH is the assessment of the subjective quality of macroscopic holograms, since for example changes in the speckle patterns are barely noticeable with the naked eye but result in vastly different holograms in terms of common quality measures such as mean-squared error-based candidates or the structural similarity index measure [61]. Furthermore, those measures tend to be

very sensitive to the propagation distance of the hologram. As we have seen in Sections 4, 5.3, and 5.4 PSR representations do not fundamentally change under propagation and are very suitable to identify several types of distortions. If applied to the full 2D holograms they allow comparisons in the full 4D phase space. This approach may yield in the near future a similarity measure for DH. Research on this matter is currently ongoing.

### 5.9. Design and Understanding of Limitations of Digital Holography Setups

By the argument presented in Section 5.2, and very much in the spirit of works in Fourier and phase space optics [13,14], it becomes possible to optimize DH setups for best bandwidth usage and therefore best quality. Exemplary stands [47], discussing the capture and display of holograms, as well as the aliasing-zones, and resolution estimates by using the WV PSR within the limits of a single object hologram in the paraxial approximation. Especially intuitive is the inclusion of sets of virtual PSFs cast from the eye(s) onto the hologram plane. Their intersection in phase space with the holographic signal marks the actually observed content, see Figure 14. Within the paraxial approximation, the "PSF of an eye" is the collection of linear chirps not unlike, Figure 9, with a slope given by $(\lambda z_o)^{-1}$. $z_o$ thereby signifies the distance from the hologram plane, which is $< 0$ whenever virtual images are observed, cf. Figure 2.

A string of consecutive articles deepened the analysis [32,62] and included design of optimized hologram representations, as well as an analysis of spatial versus angular spatial light modulator multiplexing for wide view holographic displays under the use of binocular eye PSFs [33,34]. With the help of the proposed PSRs, this work becomes now visually accessible in any scenario—including non-paraxial cases—and might be used for a more distinctive analysis of compression artifacts, see Section 5.4.3.

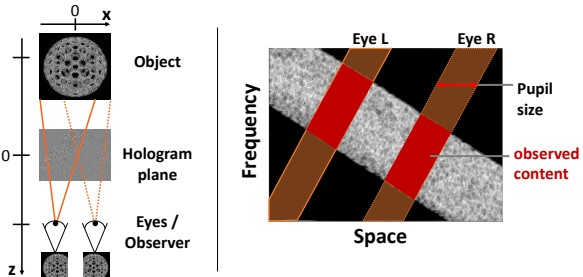

**Figure 14.** The concept of binocular vision and eye PSFs in phase space.

## 6. Conclusions

We provided a motivation of phase space analysis for DH and discussed aspects of it on individual PSFs. Thereafter, visually interpretable PSRs were discussed where we discouraged the use of the WV PSR due to the rise of cross-terms and of the spectrogram due to low-resolution for this purpose. Instead we recommended the S-method. The main part of the paper was subsequently devoted to discussing various applications of visual phase space analysis by applying the S-method to DH for the first time. In conclusion of our work, we can state that visually interpretable PSRs allow to: analyze the composition of complicated scenes without a-priori information; optimize the bandwidth usage of any DH processing chain, through analysis of the resulting DH; estimate coarsely the depth and position of objects in scene space; analyze numerical artifacts or the effects of operations such as quantization or compression; track the most important optical or scene manipulations directly in phase space. We also briefly outlined how visual understanding gained through PSRs, can be used to segment and motion compensate multi-object holograms or aid quality assessment of DH. Through the abstraction that PSRs provide, they have great potential in growing the understanding of specific holograms or holographic setups.

**Author Contributions:** Conceptualization, T.B. and T.K.; formal analysis, T.B; methodology, T.B.; software, T.B.; validation, T.B., T.K., and P.S.; data curation, T.B. and T.K; writing—original draft preparation, T.B.;

writing—review and editing, T.B., T.K., and P.S.; visualization, T.B.; supervision, T.K.; project administration, P.S.; funding acquisition, P.S. All authors have read and agreed to the published version of the manuscript.

**Funding:** The research leading to these results has received funding from the European Research Council under the European Union's Seventh Framework Programme (FP7/2007–2013)/ERC Grant Agreement Nr. 617779 (INTERFERE).

**Conflicts of Interest:** The authors declare no conflict of interest.

## Abbreviations

The following abbreviations are used in this manuscript:

| | |
|---|---|
| DH | Digital holography / digital hologram |
| H.265/HEVC | High Efficiency Video Coding (name of a video compression standard) |
| PSF | point-spread function |
| SBP | space-bandwidth product |
| SLM | spatial light modulator |
| STFT | short-term Fourier transform |
| PSR | phase space representation / time-frequency / space-spatial frequency representation |
| WV | Wigner–Ville |

## Appendix A. Estimating Space-Frequency Bandwidth Utilization

In order to estimate scene complexity with Equation (16), one needs to give a numerical estimator for $SPB_H$. Two approaches come to mind. First, one can follow the visual approach and binarize the phase space signal using a small $\varepsilon > 0$, evaluating area larger than the threshold. This approach is robust in the sense that, the area will vary only by a margin, due to resampling, for operations such as scene space rotations, translations (provided unitary propagations are used), superposition/interference, de-/magnification, changes to the surface roughness, and perspective projection. The downside is, that the approach is sensitive to the dynamic range of $H$ and hence during comparisons of holograms, their dynamic range needs to be normalized. The method is specific to the chosen parameters of the PSR, as well as the value of $\varepsilon$. Results are not transferable otherwise.

A second approach foresees the calculation of the Shannon entropy of a spectrogram or the S-method [63], which remains signal scale invariant as long all PSR parameters are kept fixed. For some non-stationary signals one might consider also the Rényi entropy paired with any PSR from Cohen's class [64]. At least in the case of the WV PSR, the Rényi entropy might not exist for arbitrary DHs and might remain too sensitive to phase differences of neighboring PSF or resampling artifacts.

## Appendix B. Estimating Depth through Linear Curve Fitting—An Explicit Example

We will demonstrate the depth estimation of the scene center on the example of Figure 9c using 1D phase space analysis and linear curve fitting. Since we evaluated the PSR on a horizontal 1D cross-section of the hologram, we will utilize the $\xi, f_\xi$ domain for our estimation. For convenience, we restate all parameters in use. The square hologram of size $N = 8192$, had a pixel pitch of $p = 1$ μm, and was recorded with $\lambda = 532$ nm light.

The S-method was evaluated with a spatial Hann window $w$ of length $M = 256$, of maximal overlap of 255 px, and a frequency filter window $v$ of length $L = 5$, which was defined in Fourier space as a Hann window. The resulting phase space image was of size $F \times S$, thus allowing to resolve $F = 256$ distinct frequency and $S = 7937$ distinct spatial channels. The origin of pixel coordinates within the PSR image is chosen to be in the top-left corner.

The scene center is defined by the central ridge of the phase space footprint of the signal. Using standard signal processing we extract two points on the linear fit that central ridge.

The obtained points have the following interpolated coordinates $(\xi_1^{px}, f_{\xi,1}^{px}) = (1\,px, 20.30\,px)$ and $(\xi_2^{px}, f_{\xi,2}^{px}) = (7937\,px, 243.77\,px)$. These coordinates are converted to $\xi$ in m and $f_\xi$ in m$^{-1}$ via

$$\xi = g(\xi^{px}) := \xi^{px}\frac{Np}{S\lambda} - \frac{Np}{2\lambda} \quad \text{and} \quad f_\xi = h(f_\xi^{px}) := \frac{f_\xi^{px} - \frac{F}{2}}{pM}\,. \tag{A1}$$

Although the Fresnel number of the hologram at the scene center is $\sim$2232, the Fresnel approximation is sufficient to estimate depth, as the pixel pitch is sufficiently large as to induce a strong band-limitation revealing only the approximately linear part around $f_\xi = 0$ of the sigmoid of the spherical PSF, compare with Figure 3c. As $z$ influences only the slope of the linear instantaneous frequency function we choose the following ansatz:

$$\frac{-1}{\lambda z} \overset{!}{=} \frac{f_{\xi,2} - f_{\xi,1}}{\xi_2 - \xi_1}\,, \tag{A2}$$

where the left-hand side is obtained by differentiation of Equation (7) after $\xi$. The right hand side is the slope of the linear fit in phase space. Combining Equations (A1) and (A2) we can estimate $z$ as

$$z = -\frac{\xi_2 - \xi_1}{\lambda\left(f_{\xi,2} - f_{\xi,1}\right)} \approx \frac{g\left(\xi_2^{px} - \xi_1^{px}\right)}{\lambda h\left(f_{\xi,2}^{px} - f_{\xi,1}^{px}\right)} \approx 1.48\,\text{cm}\,. \tag{A3}$$

The error of the obtained depth can be estimated using the total differential for our estimator of $z$ and the standard-deviation of the polynomial fit. Accounting for the width of the spatial window $v$, we actually obtain $z = 1.48 \pm 0.19$ cm which is in good agreement with the correct $z = 1.41$ cm. For larger resolutions the error decreases fast. For multi-object holograms the ridge estimates get more complicated, but holographic scene segmentation [30] prior to the analysis can help. On a final note, also the lateral position along $x$ of the scene center wrt. the optical axis can be determined with higher precision via Equation (A1), the linear phase space fit for $f_\xi$, and the ansatz $f_\xi \overset{!}{=} 0$. If the same is repeated for a vertical cross-section through the middle of the hologram, an estimate of the full 3D position of the center of the object can be obtained.

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
