# Peer review of "Providing a Visual Understanding of Holography Through Phase Space Representations"

_applsci, doi:10.3390/app10144766_

Round 1
Reviewer 1 Report
In this article the authors discuss the advantages of using the S method as a time-frequency representation for the study of Digital Holograms. The article is extensive developing in a very didactic way the different concepts. Once the choice of the S-method among others is explained the authors discuss various applications of visual phase space analysis for Digital Holograms. I strongly recommend this article for publication. Very few comments:
- Figure 1. must be explained a little bit. At least the labels of the axes of (b). e.g. What’s “Re(g)”? It seems real part of a signal function. But it should be explained,
- Line 40: A reference for “the Wigner-Ville (WV) distribution” and the “short-term Fourier transform” should be added.
- Figure 5: The labels of the axes of figure 5(a) are too big. I will also label the y axis by w (a.u.), since the dependence of m is implicitly taken into account in the plot.
- Before equation (12): Hann window. A reference here is needed.
- The use of units should follow the scientific convention along the whole work: For instance: 633nm should be changed by 633 nm.
- The same occurs with the expression of errors: 1.48cm ±19cm should be changed by: 1.48 ± 0.19 cm
Author Response
Dear reviewer,
please find our elaborate resonse letter along with the redlined version of the paper (end of the PDF) in the PDF attached.
With best regards
T. Birnbaum

Reviewer 2 Report
Title: A Visual Understanding of Holography Through Time-Frequency Representations
The proposed idea is interesting, and this is also the reason that I agreed to review this manuscript. However, I have to say that I can only glance through this manuscript. I have no time to understand every aspects of this 26 pages manuscript. This manuscript is written like a thesis or a book chapter. For example, it mentioned too many background, including section 1 and section 2 (6 pages!). The titles “3.2 What is spectrogram?” “3.3 What is S-method?” “What is the type of hologram” (line 303)…. are rarely adopted in journal papers. I understand that the authors want to inspire the readers by the lengthy explanation. However, most readers do not want to spend a lot of time reading a new thing that they do not know what to do. I suggest that the manuscript could be shorten to at least half the present length, best no more than ten pages. The authors do not need to say everything. They only need to explain their method and emphasize the advantage of their new method. Other content, if the authors think inspirable, can be included as an appendix or a supported manuscript. Alternatively, it can be submitted as an independent paper if the work is different and important. Finally, I do not like to apply the term “Time-Frequency Representations” in the field of Fourier optics because only space and spatial frequency are analyzed.
Author Response

(The authors gave the same response as above.)
